# Taming the Wild: A Unified Analysis of HOGWILD!-Style Algorithms

**Christopher De Sa, Ce Zhang, Kunle Olukotun, and Christopher Ré**
cdesa@stanford.edu, czhang@cs.wisc.edu,
kunle@stanford.edu, chrismre@stanford.edu
Departments of Electrical Engineering and Computer Science
Stanford University, Stanford, CA 94309

## Abstract

Stochastic gradient descent (SGD) is a ubiquitous algorithm for a variety of machine learning problems. Researchers and industry have developed several techniques to optimize SGD's runtime performance, including asynchronous execution and reduced precision. Our main result is a martingale-based analysis that enables us to capture the rich noise models that may arise from such techniques. Specifically, we use our new analysis in three ways: (1) we derive convergence rates for the convex case (HOGWILD!) with relaxed assumptions on the sparsity of the problem; (2) we analyze asynchronous SGD algorithms for non-convex matrix problems including matrix completion; and (3) we design and analyze an asynchronous SGD algorithm, called BUCKWILD!, that uses lower-precision arithmetic. We show experimentally that our algorithms run efficiently for a variety of problems on modern hardware.

## 1 Introduction

Many problems in machine learning can be written as a stochastic optimization problem

$$\text{minimize} \quad \mathbf{E}[\tilde{f}(x)] \quad \text{over} \quad x \in \mathbb{R}^n,$$

where $\tilde{f}$ is a random objective function. One popular method to solve this is with stochastic gradient descent (SGD), an iterative method which, at each timestep $t$, chooses a random objective sample $\tilde{f}_t$ and updates

$$x_{t+1} = x_t - \alpha \nabla \tilde{f}_t(x_t), \tag{1}$$

where $\alpha$ is the step size. For most problems, this update step is easy to compute, and perhaps because of this SGD is a ubiquitous algorithm with a wide range of applications in machine learning [1], including neural network backpropagation [2, 3, 13], recommendation systems [8, 19], and optimization [20]. For non-convex problems, SGD is popular—in particular, it is widely used in deep learning—but its success is poorly understood theoretically.

Given SGD's success in industry, practitioners have developed methods to speed up its computation. One popular method to speed up SGD and related algorithms is using asynchronous execution. In an asynchronous algorithm, such as HOGWILD! [17], multiple threads run an update rule such as Equation 1 in parallel without locks. HOGWILD! and other lock-free algorithms have been applied to a variety of uses, including PageRank approximations (FrogWild! [16]), deep learning (Dogwild! [18]) and recommender systems [24]. Many asynchronous versions of other stochastic algorithms have been individually analyzed, such as stochastic coordinate descent (SGD) [14, 15] and accelerated parallel proximal coordinate descent (APPROX) [6], producing rate results that are similar to those of HOGWILD! Recently, Gupta et al. [9] gave an empirical analysis of the effects of a low-precision variant of SGD on neural network training. Other variants of stochastic algorithms

have been proposed [5, 11, 12, 21, 22, 23]; only a fraction of these algorithms have been analyzed in the asynchronous case. Unfortunately, a new variant of SGD (or a related algorithm) may violate the assumptions of existing analysis, and hence there are gaps in our understanding of these techniques.

One approach to filling this gap is to analyze each purpose-built extension from scratch: an entirely new model for each type of asynchrony, each type of precision, etc. In a practical sense, this may be unavoidable, but ideally there would be a single technique that could analyze many models. In this vein, we prove a martingale-based result that enables us to treat many different extensions as different forms of noise within a unified model. We demonstrate our technique with three results:

1. For the convex case, HOGWILD! requires strict sparsity assumptions. Using our techniques, we are able to relax these assumptions and still derive convergence rates. Moreover, under HOGWILD!'s stricter assumptions, we recover the previous convergence rates.

2. We derive convergence results for an asynchronous SGD algorithm for a non-convex matrix completion problem. We derive the first rates for asynchronous SGD following the recent (synchronous) non-convex SGD work of De Sa et al. [4].

3. We derive convergence rates in the presence of quantization errors such as those introduced by fixed-point arithmetic. We validate our results experimentally, and show that BUCKWILD! can achieve speedups of up to $2.3\times$ over HOGWILD!-based algorithms for logistic regression.

One can combine these different methods both theoretically and empirically. We begin with our main result, which describes our martingale-based approach and our model.

## 2   Main Result

Analyzing asynchronous algorithms is challenging because, unlike in the sequential case where there is a single copy of the iterate $x$, in the asynchronous case each core has a separate copy of $x$ in its own cache. Writes from one core may take some time to be propagated to another core's copy of $x$, which results in race conditions where stale data is used to compute the gradient updates. This difficulty is compounded in the non-convex case, where a series of unlucky random events—bad initialization, inauspicious steps, and race conditions—can cause the algorithm to get stuck near a saddle point or in a local minimum.

Broadly, we analyze algorithms that repeatedly update $x$ by running an update step

$$x_{t+1} = x_t - \tilde{G}_t(x_t), \tag{2}$$

for some i.i.d. update function $\tilde{G}_t$. For example, for SGD, we would have $G(x) = \alpha \nabla \tilde{f}_t(x)$. The goal of the algorithm must be to produce an iterate in some *success region* $S$—for example, a ball centered at the optimum $x^*$. For any $T$, after running the algorithm for $T$ timesteps, we say that the algorithm has *succeeded* if $x_t \in S$ for some $t \leq T$; otherwise, we say that the algorithm has *failed*, and we denote this failure event as $F_T$.

Our main result is a technique that allows us to bound the convergence rates of asynchronous SGD and related algorithms, even for some non-convex problems. We use martingale methods, which have produced elegant convergence rate results for both convex and some non-convex [4] algorithms. Martingales enable us to model multiple forms of error—for example, from stochastic sampling, random initialization, and asynchronous delays—within a single statistical model. Compared to standard techniques, they also allow us to analyze algorithms that sometimes get stuck, which is useful for non-convex problems. Our core contribution is that a martingale-based proof for the convergence of a sequential stochastic algorithm can be easily modified to give a convergence rate for an asynchronous version.

A *supermartingale* [7] is a stochastic process $W_t$ such that $\mathbf{E}[W_{t+1}|W_t] \leq W_t$. That is, the expected value is non-increasing over time. A martingale-based proof of convergence for the sequential version of this algorithm must construct a supermartingale $W_t(x_t, x_{t-1}, \ldots, x_0)$ that is a function of both the time and the current and past iterates; this function informally represents how unhappy we are with the current state of the algorithm. Typically, it will have the following properties.

**Definition 1.** For a stochastic algorithm as described above, a non-negative process $W_t : \mathbb{R}^{n \times t} \to \mathbb{R}$ is a *rate supermartingale* with horizon $B$ if the following conditions are true. First, it must be a

supermartingale; that is, for any sequence $x_t, \ldots, x_0$ and any $t \leq B$,

$$\mathbf{E}[W_{t+1}(x_t - \tilde{G}_t(x_t), x_t, \ldots, x_0)] \leq W_t(x_t, x_{t-1}, \ldots, x_0). \tag{3}$$

Second, for all times $T \leq B$ and for any sequence $x_T, \ldots, x_0$, if the algorithm has not succeeded by time $T$ (that is, $x_t \notin S$ for all $t < T$), it must hold that

$$W_T(x_T, x_{T-1}, \ldots, x_0) \geq T. \tag{4}$$

This represents the fact that we are unhappy with running for many iterations without success.

Using this, we can easily bound the convergence rate of the sequential version of the algorithm.

**Statement 1.** *Assume that we run a sequential stochastic algorithm, for which $W$ is a* rate *super-martingale. For any $T \leq B$, the probability that the algorithm has not succeeded by time $T$ is*

$$P\left(F_T\right) \leq \frac{\mathbf{E}[W_0(x_0)]}{T}.$$

*Proof.* In what follows, we let $W_t$ denote the actual value taken on by the function in a process defined by (2). That is, $W_t = W_t(x_t, x_{t-1}, \ldots, x_0)$. By applying (3) recursively, for any $T$,

$$\mathbf{E}[W_T] \leq \mathbf{E}[W_0] = \mathbf{E}[W_0(x_0)].$$

By the law of total expectation applied to the failure event $F_T$,

$$\mathbf{E}[W_0(x_0)] \geq \mathbf{E}[W_T] = P\left(F_T\right)\mathbf{E}[W_T|F_T] + P\left(\neg F_T\right)\mathbf{E}[W_T|\neg F_T].$$

Applying (4), i.e. $\mathbf{E}[W_T|F_T] \geq T$, and recalling that $W$ is nonnegative results in

$$\mathbf{E}[W_0(x_0)] \geq P\left(F_T\right)T;$$

rearranging terms produces the result in Statement 1. □

This technique is very general; in subsequent sections we show that rate supermartingales can be constructed for SGD on all convex problems and for some algorithms for non-convex problems.

## 2.1 Modeling Asynchronicity

The behavior of an asynchronous SGD algorithm depends both on the problem it is trying to solve and on the hardware it is running on. For ease of analysis, we assume that the hardware has the following characteristics. These are basically the same assumptions used to prove the original HOG-WILD! result [17].

- There are multiple threads running iterations of (2), each with their own cache. At any point in time, these caches may hold different values for the variable $x$, and they communicate via some cache coherency protocol.

- There exists a central store $S$ (typically RAM) at which all writes are serialized. This provides a consistent value for the state of the system at any point in real time.

- If a thread performs a read $R$ of a previously written value $X$, and then writes another value $Y$ (dependent on $R$), then the write that produced $X$ will be committed to $S$ before the write that produced $Y$.

- Each write from an iteration of (2) is to only a single entry of $x$ and is done using an atomic read-add-write instruction. That is, there are no write-after-write races (handling these is possible, but complicates the analysis).

Notice that, if we let $x_t$ denote the value of the vector $x$ in the central store $S$ after $t$ writes have occurred, then since the writes are atomic, the value of $x_{t+1}$ is solely dependent on the single thread that produces the write that is serialized next in $S$. If we let $\tilde{G}_t$ denote the update function sample that is used by that thread for that write, and $v_t$ denote the cached value of $x$ used by that write, then

$$x_{t+1} = x_t - \tilde{G}_t(\tilde{v}_t) \tag{5}$$

Our hardware model further constrains the value of $\tilde{v}_t$: all the read elements of $\tilde{v}_t$ must have been written to $\mathcal{S}$ at some time before $t$. Therefore, for some nonnegative variable $\tilde{\tau}_{i,t}$,

$$e_i^T \tilde{v}_t = e_i^T x_{t-\tilde{\tau}_{i,t}},\tag{6}$$

where $e_i$ is the $i$th standard basis vector. We can think of $\tilde{\tau}_{i,t}$ as the *delay* in the $i$th coordinate caused by the parallel updates.

We can conceive of this system as a stochastic process with two sources of randomness: the noisy update function samples $\tilde{G}_t$ and the delays $\tilde{\tau}_{i,t}$. We assume that the $\tilde{G}_t$ are independent and identically distributed—this is reasonable because they are sampled independently by the updating threads. It would be unreasonable, though, to assume the same for the $\tilde{\tau}_{i,t}$, since delays may very well be correlated in the system. Instead, we assume that the delays are bounded from above by some random variable $\tilde{\tau}$. Specifically, if $\mathcal{F}_t$, the *filtration*, denotes all random events that occurred before timestep $t$, then for any $i$, $t$, and $k$,

$$\mathbf{P}\left(\tilde{\tau}_{i,t} \geq k | \mathcal{F}_t\right) \leq P\left(\tilde{\tau} \geq k\right).\tag{7}$$

We let $\tau = \mathbf{E}[\tilde{\tau}]$, and call $\tau$ the *worst-case expected delay*.

## 2.2 Convergence Rates for Asynchronous SGD

Now that we are equipped with a stochastic model for the asynchronous SGD algorithm, we show how we can use a rate supermartingale to give a convergence rate for asynchronous algorithms. To do this, we need some continuity and boundedness assumptions; we collect these into a definition, and then state the theorem.

**Definition 2.** An algorithm with rate supermartingale $W$ is $(H, R, \xi)$-bounded if the following conditions hold. First, $W$ must be Lipschitz continuous in the current iterate with parameter $H$; that is, for any $t$, $u$, $v$, and sequence $x_t, \ldots, x_0$,

$$\|W_t(u, x_{t-1}, \ldots, x_0) - W_t(v, x_{t-1}, \ldots, x_0)\| \leq H\|u - v\|.\tag{8}$$

Second, $\tilde{G}$ must be Lipschitz continuous in expectation with parameter $R$; that is, for any $u$, and $v$,

$$\mathbf{E}[\|\tilde{G}(u) - \tilde{G}(v)\|] \leq R\|u - v\|_1.\tag{9}$$

Third, the expected magnitude of the update must be bounded by $\xi$. That is, for any $x$,

$$\mathbf{E}[\|\tilde{G}(x)\|] \leq \xi.\tag{10}$$

**Theorem 1.** *Assume that we run an asynchronous stochastic algorithm with the above hardware model, for which $W$ is a $(H, R, \xi)$-bounded rate supermartingale with horizon $B$. Further assume that $HR\xi\tau < 1$. For any $T \leq B$, the probability that the algorithm has not succeeded by time $T$ is*

$$P\left(F_T\right) \leq \frac{\mathbf{E}[W(0, x_0)]}{(1 - HR\xi\tau)T}.$$

Note that this rate depends only on the worst-case expected delay $\tau$ and not on any other properties of the hardware model. Compared to the result of Statement 1, the probability of failure has only increased by a factor of $1 - HR\xi\tau$. In most practical cases, $HR\xi\tau \ll 1$, so this increase in probability is negligible.

Since the proof of this theorem is simple, but uses non-standard techniques, we outline it here. First, notice that the process $W_t$, which was a supermartingale in the sequential case, is not in the asynchronous case because of the delayed updates. Our strategy is to use $W$ to produce a new process $V_t$ that is a supermartingale in this case. For any $t$ and $x.$, if $x_u \notin S$ for all $u < t$, we define

$$V_t(x_t, \ldots, x_0) = W_t(x_t, \ldots, x_0) - HR\xi\tau t + HR \sum_{k=1}^{\infty} \|x_{t-k+1} - x_{t-k}\| \sum_{m=k}^{\infty} P\left(\tilde{\tau} \geq m\right).$$

Compared with $W$, there are two additional terms here. The first term is negative, and cancels out some of the unhappiness from (4) that we ascribed to running for many iterations. We can interpret this as us accepting that we may need to run for more iterations than in the sequential case. The second term measures the distance between recent iterates; we would be unhappy if this becomes large because then the noise from the delayed updates would also be large. On the other hand, if $x_u \in S$ for some $u < t$, then we define

$$V_t(x_t, \ldots, x_u, \ldots, x_0) = V_u(x_u, \ldots, x_0).$$

We call $V_t$ a *stopped process* because its value doesn't change after success occurs. It is straightforward to show that $V_t$ is a supermartingale for the asynchronous algorithm. Once we know this, the same logic used in the proof of Statement 1 can be used to prove Theorem 1.

Theorem 1 gives us a straightforward way of bounding the convergence time of any asynchronous stochastic algorithm. First, we find a rate supermartingale for the problem; this is typically no harder than proving sequential convergence. Second, we find parameters such that the problem is $(H, R, \xi)$-bounded, typically ; this is easily done for well-behaved problems by using differentiation to bound the Lipschitz constants. Third, we apply Theorem 1 to get a rate for asynchronous SGD. Using this method, analyzing an asynchronous algorithm is really no more difficult than analyzing its sequential analog.

## 3    Applications

Now that we have proved our main result, we turn our attention to applications. We show, for a couple of algorithms, how to construct a rate supermartingale. We demonstrate that doing this allows us to recover known rates for HOGWILD! algorithms as well as analyze cases where no known rates exist.

### 3.1    Convex Case, High Precision Arithmetic

First, we consider the simple case of using asynchronous SGD to minimize a convex function $f(x)$ using unbiased gradient samples $\nabla \tilde{f}(x)$. That is, we run the update rule

$$x_{t+1} = x_t - \alpha \nabla \tilde{f}_t(x). \tag{11}$$

We make the standard assumption that $f$ is strongly convex with parameter $c$; that is, for all $x$ and $y$

$$(x - y)^T \left( \nabla f(x) - \nabla f(y) \right) \geq c\|x - y\|^2. \tag{12}$$

We also assume continuous differentiability of $\nabla \tilde{f}$ with 1-norm Lipschitz constant $L$,

$$\mathbf{E}[\|\nabla \tilde{f}(x) - \nabla \tilde{f}(y)\|] \leq L\|x - y\|_1. \tag{13}$$

We require that the second moment of the gradient sample is also bounded for some $M > 0$ by

$$\mathbf{E}[\|\nabla \tilde{f}(x)\|^2] \leq M^2. \tag{14}$$

For some $\epsilon > 0$, we let the success region be

$$S = \{x | \|x - x^*\|^2 \leq \epsilon\}.$$

Under these conditions, we can construct a rate supermartingale for this algorithm.

**Lemma 1.** *There exists a $W_t$ where, if the algorithm hasn't succeeded by timestep $t$,*

$$W_t(x_t, \ldots, x_0) = \frac{\epsilon}{2\alpha c\epsilon - \alpha^2 M^2} \log \left( e \|x_t - x^*\|^2 \epsilon^{-1} \right) + t,$$

*such that $W_t$ is a rate submartingale for the above algorithm with horizon $B = \infty$. Furthermore, it is $(H, R, \xi)$-bounded with parameters: $H = 2\sqrt{\epsilon}(2\alpha c\epsilon - \alpha^2 M^2)^{-1}$, $R = \alpha L$, and $\xi = \alpha M$.*

Using this and Theorem 1 gives us a direct bound on the failure rate of convex HOGWILD! SGD.

**Corollary 1.** *Assume that we run an asynchronous version of the above SGD algorithm, where for some constant $\vartheta \in (0, 1)$ we choose step size*

$$\alpha = \frac{c\epsilon\vartheta}{M^2 + 2LM\tau\sqrt{\epsilon}}.$$

*Then for any $T$, the probability that the algorithm has not succeeded by time $T$ is*

$$P\left( F_T \right) \leq \frac{M^2 + 2LM\tau\sqrt{\epsilon}}{c^2\epsilon\vartheta T} \log \left( e \|x_0 - x^*\|^2 \epsilon^{-1} \right).$$

This result is more general than the result in Niu et al. [17]. The main differences are: that we make no assumptions about the sparsity structure of the gradient samples; and that our rate depends only on the second moment of $\tilde{G}$ and the expected value of $\tilde{\tau}$, as opposed to requiring absolute bounds on their magnitude. Under their stricter assumptions, the result of Corollary 1 recovers their rate.

## 3.2 Convex Case, Low Precision Arithmetic

One of the ways BUCKWILD! achieves high performance is by using low-precision fixed-point arithmetic. This introduces additional noise to the system in the form of *round-off error*. We consider this error to be part of the BUCKWILD! hardware model. We assume that the round-off error can be modeled by an unbiased rounding function operating on the update samples. That is, for some chosen precision factor $\kappa$, there is a random quantization function $\tilde{Q}$ such that, for any $x \in \mathbb{R}$, it holds that $\mathbf{E}[\tilde{Q}(x)] = x$, and the round-off error is bounded by $|\tilde{Q}(x) - x| < \alpha \kappa M$. Using this function, we can write a low-precision asynchronous update rule for convex SGD as

$$x_{t+1} = x_t - \tilde{Q}_t \left( \alpha \nabla \tilde{f}_t(\tilde{v}_t) \right), \tag{15}$$

where $\tilde{Q}_t$ operates only on the single nonzero entry of $\nabla \tilde{f}_t(\tilde{v}_t)$. In the same way as we did in the high-precision case, we can use these properties to construct a rate supermartingale for the low-precision version of the convex SGD algorithm, and then use Theorem 1 to bound the failure rate of convex BUCKWILD!

**Corollary 2.** *Assume that we run asynchronous low-precision convex SGD, and for some $\vartheta \in (0, 1)$, we choose step size*

$$\alpha = \frac{c\epsilon\vartheta}{M^2(1 + \kappa^2) + LM\tau(2 + \kappa^2)\sqrt{\epsilon}},$$

*then for any $T$, the probability that the algorithm has not succeeded by time $T$ is*

$$P(F_T) \leq \frac{M^2(1 + \kappa^2) + LM\tau(2 + \kappa^2)\sqrt{\epsilon}}{c^2\epsilon\vartheta T} \log\left(e \|x_0 - x^*\|^2 \epsilon^{-1}\right).$$

Typically, we choose a precision such that $\kappa \ll 1$; in this case, the increased error compared to the result of Corollary 1 will be negligible and we will converge in a number of samples that is very similar to the high-precision, sequential case. Since each BUCKWILD! update runs in less time than an equivalent HOGWILD! update, this result means that an execution of BUCKWILD! will produce same-quality output in less wall-clock time compared with HOGWILD!

## 3.3 Non-Convex Case, High Precision Arithmetic

Many machine learning problems are non-convex, but are still solved in practice with SGD. In this section, we show that our technique can be adapted to analyze non-convex problems. Unfortunately, there are no general convergence results that provide rates for SGD on non-convex problems, so it would be unreasonable to expect a general proof of convergence for non-convex HOGWILD! Instead, we focus on a particular problem, low-rank least-squares matrix completion,

$$\begin{aligned} \text{minimize} \quad & \mathbf{E}[\|\tilde{A} - xx^T\|_F^2] \\ \text{subject to} \quad & x \in \mathbb{R}^n, \end{aligned} \tag{16}$$

for which there exists a sequential SGD algorithm with a martingale-based rate that has already been proven. This problem arises in general data analysis, subspace tracking, principle component analysis, recommendation systems, and other applications [4]. In what follows, we let $A = \mathbf{E}[\tilde{A}]$. We assume that $A$ is symmetric, and has unit eigenvectors $u_1, u_2, \ldots, u_n$ with corresponding eigenvalues $\lambda_1 > \lambda_2 \geq \cdots \geq \lambda_n$. We let $\Delta$, the *eigengap*, denote $\Delta = \lambda_1 - \lambda_2$.

De Sa et al. [4] provide a martingale-based rate of convergence for a particular SGD algorithm, Alecton, running on this problem. For simplicity, we focus on only the rank-1 version of the problem, and we assume that, at each timestep, a single entry of $A$ is used as a sample. Under these conditions, Alecton uses the update rule

$$x_{t+1} = (I + \eta n^2 e_{\tilde{i}_t} e_{\tilde{i}_t}^T A e_{\tilde{j}_t} e_{\tilde{j}_t}^T) x_t, \tag{17}$$

where $\tilde{i}_t$ and $\tilde{j}_t$ are randomly-chosen indices in $[1, n]$. It initializes $x_0$ uniformly on the sphere of some radius centered at the origin. We can equivalently think of this as a stochastic power iteration algorithm. For any $\epsilon > 0$, we define the *success set $S$* to be

$$S = \{x | (u_1^T x)^2 \geq (1 - \epsilon) \|x\|^2\}. \tag{18}$$

That is, we are only concerned with the direction of $x$, not its magnitude; this algorithm only recovers the dominant eigenvector of $A$, not its eigenvalue. In order to show convergence for this entrywise sampling scheme, De Sa et al. [4] require that the matrix $A$ satisfy a *coherence bound* [10].

Table 1: Training loss of SGD as a function of arithmetic precision for logistic regression.

| Dataset | Rows | Columns | Size | 32-bit float | 16-bit int | 8-bit int |
|---|---|---|---|---|---|---|
| Reuters | 8K | 18K | 1.2GB | 0.5700 | 0.5700 | 0.5709 |
| Forest | 581K | 54 | 0.2GB | 0.6463 | 0.6463 | 0.6447 |
| RCV1 | 781K | 47K | 0.9GB | 0.1888 | 0.1888 | 0.1879 |
| Music | 515K | 91 | 0.7GB | 0.8785 | 0.8785 | 0.8781 |

**Definition 3.** A matrix $A \in \mathbb{R}^{n \times n}$ is incoherent with parameter $\mu$ if for every standard basis vector $e_j$, and for all unit eigenvectors $u_i$ of the matrix, $(e_j^T u_i)^2 \leq \mu^2 n^{-1}$.

They also require that the step size be set, for some constants $0 < \gamma \leq 1$ and $0 < \vartheta < (1 + \epsilon)^{-1}$ as

$$\eta = \frac{\Delta \epsilon \gamma \vartheta}{2n\mu^4 \|A\|_F^2}.$$

For ease of analysis, we add the additional assumptions that our algorithm runs in some bounded space. That is, for some constant $C$, at all times $t$, $1 \leq \|x_t\|$ and $\|x_t\|_1 \leq C$. As in the convex case, by following the martingale-based approach of De Sa et al. [4], we are able to generate a rate supermartingale for this algorithm—to save space, we only state its initial value and not the full expression.

**Lemma 2.** *For the problem above, choose any horizon $B$ such that $\eta \gamma \epsilon \Delta B \leq 1$. Then there exists a function $W_t$ such that $W_t$ is a rate supermartingale for the above non-convex SGD algorithm with parameters $H = 8n\eta^{-1}\gamma^{-1}\Delta^{-1}\epsilon^{-\frac{1}{2}}$, $R = \eta\mu \|A\|_F$, and $\xi = \eta\mu \|A\|_F C$, and*

$$\mathbf{E}\left[W_0(x_0)\right] \leq 2\eta^{-1}\Delta^{-1} \log(en\gamma^{-1}\epsilon^{-1}) + B\sqrt{2\pi\gamma}.$$

Note that the analysis parameter $\gamma$ allows us to trade off between $B$, which determines how long we can run the algorithm, and the initial value of the supermartingale $\mathbf{E}\left[W_0(x_0)\right]$. We can now produce a corollary about the convergence rate by applying Theorem 1 and setting $B$ and $T$ appropriately.

**Corollary 3.** *Assume that we run* HOGWILD! *Alecton under these conditions for $T$ timesteps, as defined below. Then the probability of failure, $P(F_T)$, will be bounded as below.*

$$T = \frac{4n\mu^4 \|A\|_F^2}{\Delta^2 \epsilon \gamma \vartheta \sqrt{2\pi\gamma}} \log\left(\frac{en}{\gamma\epsilon}\right), \qquad P(F_T) \leq \frac{\sqrt{8\pi\gamma}\mu^2}{\mu^2 - 4C\vartheta\tau\sqrt{\epsilon}}.$$

The fact that we are able to use our technique to analyze a non-convex algorithm illustrates its generality. Note that it is possible to combine our results to analyze asynchronous low-precision non-convex SGD, but the resulting formulas are complex, so we do not include them here.

## 4 Experiments

We validate our theoretical results for both asynchronous non-convex matrix completion and BUCK-WILD!, a HOGWILD! implementation with lower-precision arithmetic. Like HOGWILD!, a BUCK-WILD! algorithm has multiple threads running an update rule (2) in parallel without locking. Compared with HOGWILD!, which uses 32-bit floating point numbers to represent input data, BUCK-WILD! uses limited-precision arithmetic by rounding the input data to 8-bit or 16-bit integers. This not only decreases the memory usage, but also allows us to take advantage of single-instruction-multiple-data (SIMD) instructions for integers on modern CPUs.

We verified our main claims by running HOGWILD! and BUCKWILD! algorithms on the discussed applications. Table 1 shows how the training loss of SGD for logistic regression, a convex problem, varies as the precision is changed. We ran SGD with step size $\alpha = 0.0001$; however, results are similar across a range of step sizes. We analyzed all four datasets reported in DimmWitted [25] that favored HOGWILD!: Reuters and RCV1, which are text classification datasets; Forest, which arises from remote sensing; and Music, which is a music classification dataset. We implemented all GLM models reported in DimmWitted, including SVM, Linear Regression, and Logistic Regression, and

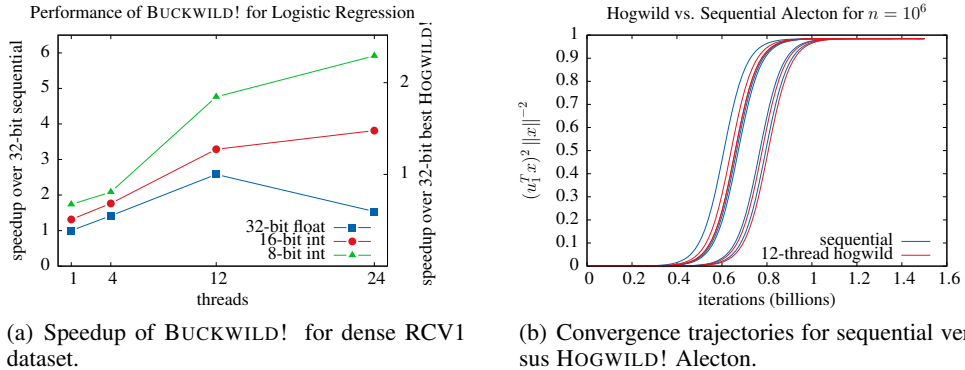

(a) Speedup of BUCKWILD! for dense RCV1 dataset.

(b) Convergence trajectories for sequential versus HOGWILD! Alecton.

Figure 1: Experiments compare the training loss, performance, and convergence of HOGWILD! and BUCKWILD! algorithms with sequential and/or high-precision versions.

report Logistic Regression because other models have similar performance. The results illustrate that there is almost no increase in training loss as the precision is decreased for these problems. We also investigated 4-bit and 1-bit computation: the former was slower than 8-bit due to a lack of 4-bit SIMD instructions, and the latter discarded too much information to produce good quality results.

Figure 1(a) displays the speedup of BUCKWILD! running on the dense-version of the RCV1 dataset compared to both full-precision sequential SGD (left axis) and best-case HOGWILD! (right axis). Experiments ran on a machine with two Xeon X650 CPUs, each with six hyperthreaded cores, and 24GB of RAM. This plot illustrates that incorporating low-precision arithmetic into our algorithm allows us to achieve significant speedups over both sequential and HOGWILD! SGD. (Note that we don't get full linear speedup because we are bound by the available memory bandwidth; beyond this limit, adding additional threads provides no benefits while increasing conflicts and thrashing the L1 and L2 caches.) This result, combined with the data in Table 1, suggest that by doing low-precision asynchronous updates, we can get speedups of up to $2.3\times$ on these sorts of datasets without a significant increase in error.

Figure 1(b) compares the convergence trajectories of HOGWILD! and sequential versions of the non-convex Alecton matrix completion algorithm on a synthetic data matrix $A \in \mathbb{R}^{n \times n}$ with ten random eigenvalues $\lambda_i > 0$. Each plotted series represents a different run of Alecton; the trajectories differ somewhat because of the randomness of the algorithm. The plot shows that the sequential and asynchronous versions behave qualitatively similarly, and converge to the same noise floor. For this dataset, sequential Alecton took $6.86$ seconds to run while 12-thread HOGWILD! Alecton took $1.39$ seconds, a $4.9\times$ speedup.

## 5 Conclusion

This paper presented a unified theoretical framework for producing results about the convergence rates of asynchronous and low-precision random algorithms such as stochastic gradient descent. We showed how a martingale-based rate of convergence for a sequential, full-precision algorithm can be easily leveraged to give a rate for an asynchronous, low-precision version. We also introduced BUCKWILD!, a strategy for SGD that is able to take advantage of modern hardware resources for both task and data parallelism, and showed that it achieves near linear parallel speedup over sequential algorithms.

**Acknowledgments**

The BUCKWILD! name arose out of conversations with Benjamin Recht. Thanks also to Madeleine Udell for helpful conversations. The authors acknowledge the support of: DARPA FA8750-12-2-0335; NSF IIS-1247701; NSF CCF-1111943; DOE 108845; NSF CCF-1337375; DARPA FA8750-13-2-0039; NSF IIS-1353606; ONR N000141210041 and N000141310129; NIH U54EB020405; Oracle; NVIDIA; Huawei; SAP Labs; Sloan Research Fellowship; Moore Foundation; American Family Insurance; Google; and Toshiba.

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
