[Supplementary Material]

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

# A Proof of Theorem 1

*Proof of Theorem 1.* This proof is a more detailed version of the argument outlined in Section 2.2. First, we restate the definition of the process $V_t$ from the body of the paper. As long as the algorithm hasn't succeeded yet,

$$V_t(x_t, \ldots, x_0) = W_t(x_t, \ldots, x_0) - HR\xi\tau t + HR \sum_{k=1}^{\infty} \|x_{t-k+1} - x_{t-k}\| \sum_{m=k}^{\infty} P\left(\tilde{\tau} \geq m\right).$$

At the next timestep, we will have $x_{t+1} = x_t + \tilde{G}(\tilde{v}_t)$, and so

$$V_{t+1}(x_t + \tilde{G}(\tilde{v}_t), x_t, \ldots, x_0) = W_{t+1}(x_t + \tilde{G}(\tilde{v}_t), x_t, \ldots, x_0) - HR\xi\tau(t+1)$$

$$+ HR \left\|\tilde{G}(\tilde{v}_t)\right\| \sum_{m=1}^{\infty} P\left(\tilde{\tau} \geq m\right)$$

$$+ HR \sum_{k=2}^{\infty} \|x_{t-k+2} - x_{t-k+1}\| \sum_{m=k}^{\infty} P\left(\tilde{\tau} \geq m\right).$$

Re-indexing the second sum and applying the definition of $\tau$ produces

$$V_{t+1}(x_t + \tilde{G}(\tilde{v}_t), x_t, \ldots, x_0) = W_{t+1}(x_t + \tilde{G}(\tilde{v}_t), x_t, \ldots, x_0) - HR\xi\tau(t+1) + HR\tau \left\|\tilde{G}(\tilde{v}_t)\right\|$$

$$+ HR \sum_{k=1}^{\infty} \|x_{t-k+1} - x_{t-k}\| \sum_{m=k+1}^{\infty} P\left(\tilde{\tau} \geq m\right).$$

Applying the Lipschitz continuity assumption (8) for $W$ results in

$$V_{t+1}(x_t + \tilde{G}(\tilde{v}_t), x_t, \ldots, x_0) \leq W_{t+1}(x_t + \tilde{G}(x_t), x_t, \ldots, x_0) + H \left\|\tilde{G}(\tilde{v}_t) - \tilde{G}(x_t)\right\|$$

$$- HR\xi\tau(t+1) + HR\tau \left\|\tilde{G}(\tilde{v}_t)\right\|$$

$$+ HR \sum_{k=1}^{\infty} \|x_{t-k+1} - x_{t-k}\| \sum_{m=k+1}^{\infty} P\left(\tilde{\tau} \geq m\right).$$

Taking the expected value of both sides produces

$$\mathbf{E}\left[V_{t+1}(x_t + \tilde{G}(\tilde{v}_t), x_t, \ldots, x_0)\right] \leq \mathbf{E}\left[W_{t+1}(x_t + \tilde{G}(x_t), x_t, \ldots, x_0)\right] + H\mathbf{E}\left[\left\|\tilde{G}(\tilde{v}_t) - \tilde{G}(x_t)\right\|\right]$$

$$- HR\xi\tau(t+1) + HR\tau\mathbf{E}\left[\left\|\tilde{G}(\tilde{v}_t)\right\|\right]$$

$$+ HR \sum_{k=1}^{\infty} \|x_{t-k+1} - x_{t-k}\| \sum_{m=k+1}^{\infty} P\left(\tilde{\tau} \geq m\right).$$

Applying the rate supermartingale property (3) of $W$,

$$\mathbf{E}\left[V_t(x_t + \tilde{G}(\tilde{v}_t), x_t, \ldots, x_0)\right] \leq W_t(x_t, \ldots, x_0) + H\mathbf{E}\left[\left\|\tilde{G}(\tilde{v}_t) - \tilde{G}(x_t)\right\|\right]$$

$$- HR\xi\tau(t+1) + HR\tau\mathbf{E}\left[\left\|\tilde{G}(\tilde{v}_t)\right\|\right]$$

$$+ HR \sum_{k=1}^{\infty} \|x_{t-k+1} - x_{t-k}\| \sum_{m=k+1}^{\infty} P\left(\tilde{\tau} \geq m\right).$$

Applying the Lipschitz continuity assumption (9) for $\tilde{G}$,

$$\mathbf{E}\left[V_t(x_t + \tilde{G}(\tilde{v}_t), x_t, \ldots, x_0)\right] \leq W_t(x_t, \ldots, x_0) + HR\mathbf{E}\left[\|\tilde{v}_t - x_t\|_1\right]$$
$$- HR\xi\tau(t+1) + HR\tau\mathbf{E}\left[\left\|\tilde{G}(\tilde{v}_t)\right\|\right]$$
$$+ HR\sum_{k=1}^{\infty}\|x_{t-k+1} - x_{t-k}\| \sum_{m=k+1}^{\infty} P\left(\tilde{\tau} \geq m\right).$$

Finally, applying the update distance bound (10),

$$\mathbf{E}\left[V_t(x_t + \tilde{G}(\tilde{v}_t), x_t, \ldots, x_0)\right] \leq W_t(x_t, \ldots, x_0) + HR\mathbf{E}\left[\|\tilde{v}_t - x_t\|_1\right] - HR\xi\tau(t+1)$$
$$+ HR\xi\tau + HR\sum_{k=1}^{\infty}\|x_{t-k+1} - x_{t-k}\| \sum_{m=k+1}^{\infty} P\left(\tilde{\tau} \geq m\right)$$
$$= W_t(x_t, \ldots, x_0) - HR\xi\tau t$$
$$+ HR\sum_{k=1}^{\infty}\|x_{t-k+1} - x_{t-k}\| \sum_{m=k}^{\infty} P\left(\tilde{\tau} \geq m\right)$$
$$+ HR\mathbf{E}\left[\|\tilde{v}_t - x_t\|_1\right] - HR\sum_{k=1}^{\infty}\|x_{t-k+1} - x_{t-k}\| P\left(\tilde{\tau} \geq k\right)$$
$$= V_t(x_t, \ldots, x_0) + HR\mathbf{E}\left[\|\tilde{v}_t - x_t\|_1\right]$$
$$- HR\sum_{k=1}^{\infty}\|x_{t-k+1} - x_{t-k}\| P\left(\tilde{\tau} \geq k\right).$$

Now, by the definition of the $\tilde{v}_t$,

$$\|\tilde{v}_t - x_t\|_1 = \sum_{i=1}^{n}\left|e_i^T x_t - e_i^T \tilde{v}_t\right|$$
$$= \sum_{i=1}^{n}\left|e_i^T x_t - e_i^T x_{t-\tilde{\tau}_{i,t}}\right|$$
$$\leq \sum_{i=1}^{n}\sum_{k=1}^{\tilde{\tau}_{i,t}}\left|e_i^T x_{t-k+1} - e_i^T x_{t-k}\right|$$

Furthermore, using the bound on $\tilde{\tau}_{i,t}$ from (7) gives us

$$\mathbf{E}\left[\|\tilde{v}_t - x_t\|_1\right] \leq \sum_{i=1}^{n}\sum_{k=1}^{\infty}\left|e_i^T x_{t-k+1} - e_i^T x_{t-k}\right| P\left(\tilde{\tau}_{i,t} \geq k\right)$$
$$\leq \sum_{i=1}^{n}\sum_{k=1}^{\infty}\left|e_i^T x_{t-k+1} - e_i^T x_{t-k}\right| P\left(\tilde{\tau} \geq k\right)$$
$$= \sum_{k=1}^{\infty}\|x_{t-k+1} - x_{t-k}\|_1 P\left(\tilde{\tau} \geq k\right)$$
$$= \sum_{k=1}^{\infty}\|x_{t-k+1} - x_{t-k}\| P\left(\tilde{\tau} \geq k\right),$$

where the 1-norm is equal to the 2-norm here because each step only updates a single entry of $x$. Substituting this result in to the above equation allows us to conclude that, if the algorithm hasn't succeeded by time $t$,

$$\mathbf{E}\left[V_t(x_t + \tilde{G}(\tilde{v}_t), x_t, \ldots, x_0)\right] \leq V_t(x_t, \ldots, x_0). \tag{19}$$

On the other hand, if it has succeeded, this statement will be vacuously true, since $V_t$ does not change after success occurs. Therefore, (19) will hold for all times.

In what follows, as in the proof of Statement 1, we let $V_t$ denote the actual value taken on by the function during execution of the algorithm. That is, $V_t = V_t(x_t, x_{t-1}, \ldots, x_0)$. By applying (19) recursively, for any $T < B$, we can show that

$$\mathbf{E}\left[V_T\right] \leq \mathbf{E}\left[V_0\right].$$

Since we assumed as part of our hardware model that $x_t = x_0$ for $t < 0$,

$$\mathbf{E}\left[V_0\right] = \mathbf{E}\left[W_0(x_0)\right].$$

Therefore, by the law of total expectation

$$
\begin{aligned}
\mathbf{E}\left[W_0(x_0)\right] &\geq \mathbf{E}\left[V_T\right] \\
&= \mathbf{E}\left[V_T|F_T\right] P\left(F_T\right) + \mathbf{E}\left[V_T|\neg F_T\right] P\left(\neg F_T\right) \\
&\geq \mathbf{E}\left[V_T|F_T\right] P\left(F_T\right) \\
&= \mathbf{E}\left[ W_T(x_T, \ldots, x_0) - HR\xi\tau T \right. \\
&\qquad \left. + HR\sum_{k=1}^{\infty} \|x_{T-k+1} - x_{T-k}\| \sum_{m=k}^{\infty} P\left(\tilde{\tau} \geq m\right) \middle| F_T \right] P\left(F_T\right) \\
&\geq \left(\mathbf{E}\left[W_T(x_T, \ldots, x_0)|\mathcal{F}_T\right] - HR\xi\tau T\right) P\left(F_T\right).
\end{aligned}
$$

Since $W_t$ is a rate supermartingale, we can apply (4) to get

$$\mathbf{E}\left[W_0(x_0)\right] \geq \left(T - HR\xi\tau T\right) P\left(F_T\right),$$

and solving for $P\left(F_T\right)$ produces

$$P\left(F_T\right) \leq \frac{\mathbf{E}\left[W_0(x_0)\right]}{(1 - HR\xi\tau)T},$$

as desired. $\qquad\square$

## B  Proofs for Convex Case

First, we state the rate supermartingale lemma for the low-precision convex SGD algorithm.

**Lemma 3.** *There exists a $W_t$ with*

$$W_0(x_0) \leq \frac{\epsilon}{2\alpha c\epsilon - \alpha^2 M^2(1+\kappa^2)} \log\left(\frac{e\,\|x_0 - x^*\|^2}{\epsilon}\right)$$

*such that $W_t$ is a rate submartingale for the above convex SGD algorithm with horizon $B = \infty$. Furthermore, it is $(H, R, \xi)$-bounded with parameters: $R = \alpha L$, $\xi^2 = \alpha^2(1+\kappa^2)M^2$, and*

$$H = \frac{2\sqrt{\epsilon}}{2\alpha c\epsilon - \alpha^2 M^2(1+\kappa^2)}.$$

We note that, including this Lemma, the results in Section 3.1 are the same as the results in Section 3.2, except that the quantization factor is set as $\kappa = 0$. It follows that it is sufficient to prove only the Lemma and Corollary in 3.2; this is what we will do here.

In order to prove the results in this section, we will need some definitions and lemmas, which we state now.

**Definition 4** (Piecewise Logarithm)**.** For the purposes of this document, we define the *piecewise logarithm* function to be

$$\log(x) = \begin{cases} \log(ex) & : x \geq 1 \\ x & : x \leq 1 \end{cases}$$

**Lemma 4.** *The piecewise logarithm function is differentiable and concave. Also, if $x \geq 1$, then for any $\Delta$,*

$$\log(x(1+\Delta)) \leq \log(x) + \Delta.$$

*Proof.* The first part of the lemma follows from the fact that $\log(x)$ is a piecewise function, where the pieces are both increasing and concave, and the fact that the function is differentiable at $x = 1$. The second part of the lemma follows from the fact that a first-order approximation always overestimates a concave function. $\square$

Armed with this definition, we prove Lemma 3.

*Proof of Lemma 3.* First, we note that, at any timestep $t$, if we evaluate the distance to the optimum at the next timestep using (11), then

$$\left\| x_t + \tilde{G}_t(x_t) - x^* \right\|^2 = \|x_t - x^*\|^2 - 2(x_t - x^*)^T \tilde{Q}_t \left( \alpha \nabla \tilde{f}_t(x_t) \right) + \left\| \tilde{Q}_t \left( \alpha \nabla \tilde{f}_t(x_t) \right) \right\|^2$$

$$= \|x_t - x^*\|^2 - 2(x_t - x^*)^T \tilde{Q}_t \left( \alpha \nabla \tilde{f}_t(x_t) \right)$$

$$+ \alpha^2 \left\| \alpha \nabla \tilde{f}_t(x_t) \right\|^2 + \left\| \tilde{Q}_t \left( \alpha \nabla \tilde{f}_t(x_t) \right) - \alpha \nabla \tilde{f}_t(x_t) \right\|^2.$$

Taking the expected value and applying (14), and the bounds on the properties of $\tilde{Q}_t$, produces

$$\mathbf{E}\left[ \left\| x_t + \tilde{G}_t(x_t) - x^* \right\|^2 \right] \leq \|x_t - x^*\|^2 - 2\alpha(x_t - x^*)^T \nabla f(x_t) + \alpha^2 M^2 + \delta^2.$$

Since we assigned $\delta \leq \alpha \kappa M$,

$$\mathbf{E}\left[ \left\| x_t + \tilde{G}_t(x_t) - x^* \right\|^2 \right] \leq \|x_t - x^*\|^2 - 2\alpha(x_t - x^*)^T \nabla f(x_t) + \alpha^2 M^2 (1 + \kappa^2)$$

$$= \|x_t - x^*\|^2 - 2\alpha(x_t - x^*)^T \left( \nabla f(x_t) - \nabla f(x^*) \right) + \alpha^2 M^2 (1 + \kappa^2).$$

Applying the strong convexity assumption (12),

$$\mathbf{E}\left[ \left\| x_t + \tilde{G}_t(x_t) - x^* \right\|^2 \right] \leq \|x_t - x^*\|^2 - 2\alpha c \|x_t - x^*\|^2 + \alpha^2 M^2 (1 + \kappa^2)$$

$$= (1 - 2\alpha c) \|x_t - x^*\|^2 + \alpha^2 M^2 (1 + \kappa^2).$$

Now, if we haven't succeeded yet, then $\|x_t - x^*\|^2 > \epsilon$. Under these conditions,

$$\mathbf{E}\left[ \left\| x_t + \tilde{G}_t(x_t) - x^* \right\|^2 \right] \leq \|x_t - x^*\|^2 \left( 1 - 2\alpha c + \alpha^2 M^2 (1 + \kappa^2)\epsilon^{-1} \right).$$

Multiplying both sides of the equation by $\epsilon^{-1}$ and taking the piecewise logarithm, by Jensen's inequality

$$\mathbf{E}\left[ \log \left( \epsilon^{-1} \left\| x_t + \tilde{G}_t(x_t) - x^* \right\|^2 \right) \right] \leq \log \left( \mathbf{E}\left[ \epsilon^{-1} \left\| x_t + \tilde{G}_t(x_t) - x^* \right\|^2 \right] \right)$$

$$\leq \log \left( \epsilon^{-1} \|x_t - x^*\|^2 \left( 1 - 2\alpha c + \alpha^2 M^2 (1 + \kappa^2)\epsilon^{-1} \right) \right).$$

Since $\epsilon^{-1} \|x_t - x^*\|^2 > 1$, we can apply Lemma 4, which gives us

$$\mathbf{E}\left[ \log \left( \epsilon^{-1} \left\| x_t + \tilde{G}_t(x_t) - x^* \right\|^2 \right) \right] \leq \log \left( \epsilon^{-1} \|x_t - x^*\|^2 \right) - 2\alpha c + \alpha^2 M^2 (1 + \kappa^2)\epsilon^{-1}.$$

Now, we define the rate supermartingale $W_t$ such that, if we haven't succeeded up to time $t$, then

$$W_t(x_t, \ldots, x_0) = \frac{\epsilon}{2\alpha c \epsilon - \alpha^2 M^2 (1 + \kappa^2)} \log \left( \epsilon^{-1} \|x_t - x^*\|^2 \right) + t;$$

otherwise, if $u$ is a time such that $x_u \in S$, then for all $t > u$,

$$W_t(x_t, \ldots, x_0) = W_u(x_u, \ldots, x_0).$$

The first rate supermartingale property (3) is true because if success hasn't occurred,

$$\mathbf{E}\left[W_{t+1}(x_t + \tilde{G}_t(x_t), \ldots, x_0)\right] = \mathbf{E}\left[\frac{\epsilon}{2\alpha c\epsilon - \alpha^2 M^2(1 + \kappa^2)} \log\left(\epsilon^{-1}\left\|x_t + \tilde{G}_t(x_t) - x^*\right\|^2\right)\right.$$
$$\left. + (t+1)\right]$$
$$= \frac{\epsilon}{2\alpha c\epsilon - \alpha^2 M^2(1 + \kappa^2)}\mathbf{E}\left[\log\left(\epsilon^{-1}\left\|x_t + \tilde{G}_t(x_t) - x^*\right\|^2\right)\right]$$
$$+ (t+1)$$
$$\leq \frac{\epsilon}{2\alpha c\epsilon - \alpha^2 M^2(1 + \kappa^2)}\left(\log\left(\epsilon^{-1}\left\|x_t - x^*\right\|^2\right) - 2\alpha c\right.$$
$$\left. + \alpha^2 M^2(1 + \kappa^2)\epsilon^{-1}\right) + (t+1)$$
$$= \frac{\epsilon}{2\alpha c\epsilon - \alpha^2 M^2(1 + \kappa^2)}\log\left(\epsilon^{-1}\left\|x_t - x^*\right\|^2\right) - 1 + (t+1)$$
$$= W_t(x_t, \ldots, x_0);$$

it is vacuously true if success has occurred because the value of $W_t$ does not change after $x_u \in S$ for $u < t$. The second rate supermartingale property (4) holds because, if success hasn't occurred by time $T$,

$$W_T(x_T, \ldots, x_0) = \frac{\epsilon}{2\alpha c\epsilon - \alpha^2 M^2(1 + \kappa^2)}\log\left(\epsilon^{-1}\left\|x_T - x^*\right\|^2\right) + T \geq T;$$

this follows from the non-negativity of the $\log$ function for non-negative arguments.

We have now shown that $W_t$ is a rate supermartingale for this algorithm. Next, we verify that the bound on $W_0$ given in the lemma statement holds. At time 0, by the definition of the $\log$ function, since we assume that success has not occurred yet,

$$W_0(x_0) = \frac{\epsilon}{2\alpha c\epsilon - \alpha^2 M^2(1 + \kappa^2)}\log\left(\epsilon^{-1}\left\|x_0 - x^*\right\|^2\right)$$
$$= \frac{\epsilon}{2\alpha c\epsilon - \alpha^2 M^2(1 + \kappa^2)}\log\left(\frac{e\left\|x_0 - x^*\right\|^2}{\epsilon}\right);$$

this is the bound given in the lemma statement.

Next, we show that this rate supermartingale is $(H, R, \xi)$-bounded, for the values of $H$, $R$, and $\xi$ given in the lemma statement. First, for any $x$, $t$, and sequence $x_{t-1}, \ldots, x_0$,

$$\nabla_x W_t(x, x_{t-1}, \ldots, x_0) = \nabla_x \left(\frac{\epsilon}{2\alpha c\epsilon - \alpha^2 M^2(1 + \kappa^2)}\log\left(\epsilon^{-1}\left\|x - x^*\right\|^2\right)\right)$$
$$= \frac{\epsilon}{2\alpha c\epsilon - \alpha^2 M^2(1 + \kappa^2)}2\epsilon^{-1}(x - x^*)\log'\left(\epsilon^{-1}\left\|x - x^*\right\|^2\right).$$

Now, by the definition of $\log$, we can conclude that $\log'(u) = \min\left(1, u^{-1}\right)$. Therefore,

$$\nabla_x W_t(x, x_{t-1}, \ldots, x_0) = \frac{2}{2\alpha c\epsilon - \alpha^2 M^2(1 + \kappa^2)}(x - x^*)\min\left(1, \epsilon\left\|x - x^*\right\|^{-2}\right),$$

and taking the norm of both sides,

$$\nabla_x W_t(x, x_{t-1}, \ldots, x_0) = \frac{2}{2\alpha c\epsilon - \alpha^2 M^2(1 + \kappa^2)}\min\left(\left\|x - x^*\right\|, \epsilon\left\|x - x^*\right\|^{-1}\right).$$

Clearly, this expression is maximized when $\|x - x^*\|^2 = \epsilon$. Therefore,

$$\nabla_x W_t(x, x_{t-1}, \ldots, x_0) \leq \frac{2\sqrt{\epsilon}}{2\alpha c\epsilon - \alpha^2 M^2 (1 + \kappa^2)}.$$

The Lipschitz continuity expression with $H$ in the lemma statement now follows from the mean value theorem.

Next, we bound the Lipschitz continuity expression for $R$. We have that, for any $x$ and $y$, if the single non-zero entry of $\nabla \tilde{f}$ is at index $i$, then

$$\mathbf{E}\left[\left\|\tilde{G}(x) - \tilde{G}(y)\right\|\right] = \mathbf{E}\left[\left\|\tilde{Q}(\alpha \nabla \tilde{f}(x)) - \tilde{Q}(\alpha \nabla \tilde{f}(y))\right\|\right]$$
$$= \mathbf{E}\left[\left|\tilde{Q}(\alpha e_i^T \nabla \tilde{f}(x)) - \tilde{Q}(\alpha e_i^T \nabla \tilde{f}(y))\right|\right]$$

Without loss of generality, we assume that $\tilde{Q}$ is non-decreasing, and that $e_i^T \nabla \tilde{f}(x) \geq e_i^T \nabla \tilde{f}(y)$. Thus, by the unbiased quality of $\tilde{Q}$,

$$\mathbf{E}\left[\left\|\tilde{G}(x) - \tilde{G}(y)\right\|\right] = \mathbf{E}\left[\tilde{Q}(e_i^T \alpha \nabla \tilde{f}(x)) - \tilde{Q}(e_i^T \alpha \nabla \tilde{f}(y))\right]$$
$$= \mathbf{E}\left[e_i^T \alpha \nabla \tilde{f}(x) - e_i^T \alpha \nabla \tilde{f}(y)\right]$$
$$= \alpha \mathbf{E}\left[\left\|\nabla \tilde{f}(x) - \nabla \tilde{f}(y)\right\|\right].$$

Finally, applying (13),

$$\mathbf{E}\left[\left\|\tilde{G}(x) - \tilde{G}(y)\right\|\right] \leq \alpha L.$$

Finally, we bound the update expression with $\xi$. We have,

$$\mathbf{E}\left[\left\|\tilde{G}(x)\right\|\right]^2 = \mathbf{E}\left[\left\|\tilde{Q}(\alpha \nabla \tilde{f}(x))\right\|\right]^2$$
$$\leq \mathbf{E}\left[\left\|\tilde{Q}(\alpha \nabla \tilde{f}(x))\right\|^2\right]$$
$$= \mathbf{E}\left[\alpha^2 \left\|\nabla \tilde{f}(x)\right\|^2 + 2\alpha (\nabla \tilde{f}(x))^T \left(\tilde{Q}(\alpha \nabla \tilde{f}(x)) - \alpha \nabla \tilde{f}(x)\right)\right.$$
$$\left. + \left\|\tilde{Q}(\alpha \nabla \tilde{f}(x)) - \alpha \nabla \tilde{f}(x)\right\|^2\right].$$

Applying the bounds on the rounding error,

$$\mathbf{E}\left[\left\|\tilde{G}(x)\right\|\right]^2 \leq \mathbf{E}\left[\alpha^2 \left\|\nabla \tilde{f}(x)\right\|^2 + 2\alpha (\nabla \tilde{f}(x))^T \left(\tilde{Q}(\alpha \nabla \tilde{f}(x)) - \alpha \nabla \tilde{f}(x)\right) + \delta^2\right].$$

Taking the expected value and applying (14) and the unbiased quality of $\tilde{Q}$,

$$\mathbf{E}\left[\left\|\tilde{G}(x)\right\|\right]^2 \leq \alpha^2 M^2 + \delta^2.$$

Applying the assignment $\delta = \alpha \kappa M$ results in

$$\mathbf{E}\left[\left\|\tilde{G}(x)\right\|\right]^2 \leq \alpha^2 M^2 (1 + \kappa^2),$$

which is the desired expression.

So, we have proved all the statements in the lemma. □

*Proof of Corollary 2.* Applying Theorem 1 directly to the result of Lemma 1 produces

$$P\left(F_T\right) \leq \frac{\mathbf{E}\left[W_0(x_0)\right]}{(1 - HR\xi\tau)T}$$

$$= \frac{\epsilon}{2\alpha c\epsilon - \alpha^2 M^2(1 + \kappa^2)} \log\left(\frac{e\left\|x_0 - x^*\right\|^2}{\epsilon}\right)\left(\left(1\right.\right.$$

$$\left.\left. - \left(\frac{2\sqrt{\epsilon}}{2\alpha c\epsilon - \alpha^2 M^2(1 + \kappa^2)}\right)(\alpha L)(\alpha M\sqrt{1 + \kappa^2})\tau\right)T\right)^{-1}$$

$$= \frac{\epsilon}{\left(2\alpha c\epsilon - \alpha^2\left(M^2(1 + \kappa^2) - 2LM\tau\sqrt{1 + \kappa^2}\sqrt{\epsilon}\right)\right)T} \log\left(\frac{e\left\|x_0 - x^*\right\|^2}{\epsilon}\right)$$

$$\leq \frac{\epsilon}{\left(2\alpha c\epsilon - \alpha^2\left(M^2(1 + \kappa^2) - LM\tau(2 + \kappa^2)\sqrt{\epsilon}\right)\right)T} \log\left(\frac{e\left\|x_0 - x^*\right\|^2}{\epsilon}\right)$$

Substituting the chosen value of $\alpha$,

$$P\left(F_T\right) \leq \frac{\epsilon}{T}\left(2c\epsilon\left(\frac{c\epsilon\vartheta}{M^2(1 + \kappa^2) + LM\tau(2 + \kappa^2)\sqrt{\epsilon}}\right) - \left(M^2(1 + \kappa^2)\right.\right.$$

$$\left.\left. - LM\tau(2 + \kappa^2)\sqrt{\epsilon}\right)\left(\frac{c\epsilon\vartheta}{M^2(1 + \kappa^2) + LM\tau(2 + \kappa^2)\sqrt{\epsilon}}\right)^2\right)^{-1} \log\left(\frac{e\left\|x_0 - x^*\right\|^2}{\epsilon}\right)$$

$$= \frac{\epsilon}{\left(\frac{2c^2\epsilon^2\vartheta}{M^2(1+\kappa^2)+LM\tau(2+\kappa^2)\sqrt{\epsilon}} - \frac{c^2\epsilon^2\vartheta^2}{M^2(1+\kappa^2)+LM\tau(2+\kappa^2)\sqrt{\epsilon}}\right)T} \log\left(\frac{e\left\|x_0 - x^*\right\|^2}{\epsilon}\right)$$

$$\leq \frac{\epsilon}{\frac{c^2\epsilon^2\vartheta}{M^2(1+\kappa^2)+LM\tau(2+\kappa^2)\sqrt{\epsilon}}T} \log\left(\frac{e\left\|x_0 - x^*\right\|^2}{\epsilon}\right)$$

$$= \frac{M^2(1 + \kappa^2) + LM\tau(2 + \kappa^2)\sqrt{\epsilon}}{c^2\epsilon\vartheta T} \log\left(\frac{e\left\|x_0 - x^*\right\|^2}{\epsilon}\right),$$

as desired. $\qquad\square$

## C   Proofs for Non-Convex Case

In order to accomplish this proof, we make use of some definitions and lemmas that appear in De Sa et al. [4]. We state them here before proceeding to the proof.

First, we define a function

$$\tau(x) = \frac{(u_1^T x)^2}{(1 - \gamma n^{-1})(u_1^T x)^2 + \gamma n^{-1}\left\|x\right\|^2}.$$

Clearly, $0 \leq \tau(x) \leq 1$. Using this function, De Sa et al. [4] prove the following lemma. While their version of the lemma applies to higher-rank problems and multiple distributions, we state here a version that is specialized for the rank-1, entrywise sampling case we study in this paper. (This is a combination of Lemma 2 and Lemma 12 from De Sa et al. [4].)

**Lemma 5** ($\tau$-bound). *If we run the Alecton update rule using entrywise sampling under the conditions in Section 3.3, including the incoherence and step size assignment, then for any $x \notin S$,*

$$\mathbf{E}\left[\tau(x + \eta\tilde{A}x)\right] \geq \tau(x)\left(1 + \eta\Delta(1 - \tau(x))\right).$$

We also use another lemma from De Sa et al. [4]. This is a combination of their Lemmas 1 and 7.

**Lemma 6** (Expected value of $\tau(x_0)$). *If we initialize $x_0$ with a uniform random angle (as done in Alecton), then*

$$\mathbf{E}\left[1 - \tau(x_0)\right] \leq \sqrt{\frac{\pi\gamma}{2}}.$$

Now, we prove Lemma 2.

*Proof of Lemma 2.* First, if $x \notin S$, then $(u_1^T x)^2 \leq (1 - \epsilon) \|x\|^2$. Therefore,

$$
\begin{aligned}
\tau(x) &= \frac{(u_1^T x)^2}{(1 - \gamma n^{-1})(u_1^T x)^2 + \gamma n^{-1} \|x\|^2} \\
&\leq \frac{1 - \epsilon}{(1 - \gamma n^{-1})(1 - \epsilon) + \gamma n^{-1}} \\
&= \frac{1 - \epsilon}{1 - \epsilon + \gamma n^{-1} \epsilon},
\end{aligned}
$$

and so

$$
1 - \tau(x) \geq \frac{\gamma n^{-1} \epsilon}{1 - \epsilon + \gamma n^{-1} \epsilon,} > \gamma n^{-1} \epsilon.
$$

From the result of Lemma 5, for any $x \notin S$,

$$
\mathbf{E}\left[\tau(x + \eta \tilde{A} x)\right] \geq \tau(x)\left(1 + \eta \Delta(1 - \tau(x))\right).
$$

Therefore,

$$
\mathbf{E}\left[1 - \tau(x + \eta \tilde{A} x)\right] \leq (1 - \tau(x))(1 - \eta \Delta \tau(x))
$$

Therefore, by Jensen's inequality and Lemma 4, since $\gamma^{-1} n \epsilon (1 - \tau(x)) > 1$,

$$
\begin{aligned}
\mathbf{E}\left[\log\left(\gamma^{-1} n \epsilon^{-1}\left(1 - \tau(x + \eta \tilde{A} x)\right)\right)\right] &\geq \log\left(\mathbf{E}\left[\gamma^{-1} n \epsilon^{-1}\left(1 - \tau(x + \eta \tilde{A} x)\right)\right]\right) \\
&\geq \log\left(\gamma^{-1} n \epsilon^{-1}(1 - \tau(x))(1 - \eta \Delta \tau(x))\right) \\
&\geq \log\left(\gamma^{-1} n \epsilon^{-1}(1 - \tau(x))\right) - \eta \Delta \tau(x).
\end{aligned}
$$

Now, we define our rate supermartingale. First, define

$$
Z = \left\{ x \middle| \tau(x) \geq \frac{1}{2} \right\},
$$

and let $B > 0$ be any constant. Let $W_t$ be defined such that, if $x_u \notin S \cup Z$ for all $u \leq t$, then

$$
W_t(x_t, \ldots, x_0) = \frac{2}{\eta \Delta} \log\left(\gamma^{-1} n \epsilon^{-1}(1 - \tau(x_t))\right) + 2B(1 - \tau(x_t)) + t.
$$

On the other hand, if $x_u \in S \cup Z$ for some $u$, then for all $t > u$, we define

$$
W_t(x_t, \ldots, x_0) = W_u(x_u, \ldots, x_0).
$$

That is, once $x_t$ enters $S \cup Z$, the process $W$ stops changing.

We verify that $W_t$ is a rate supermartingale. First, (3) is true because, in the case that the process has stopped it is true vacuously, and in the case that it hasn't stopped (i.e. $x_i \notin S \cup Z$ for all $u \leq t$),

$$
\begin{aligned}
\mathbf{E}\left[W_{t+1}(x_t + \eta \tilde{A}_t x_t, x_t, \ldots, x_0\right] &= \mathbf{E}\left[\frac{2}{\eta \Delta} \log\left(\gamma^{-1} n \epsilon^{-1}(1 - \tau(x_t + \eta \tilde{A}_t x_t))\right)\right. \\
&\quad \left. + 2B(1 - \tau(x_t + \eta \tilde{A}_t x_t)) + t + 1\right] = \frac{2}{\eta \Delta} \mathbf{E}\left[\log\left(\gamma^{-1} n \epsilon^{-1}(1 - \tau(x_t + \eta \tilde{A}_t x_t))\right)\right] \\
&\quad + 2B \mathbf{E}\left[1 - \tau(x_t + \eta \tilde{A}_t x_t)\right] + t + 1 \leq \frac{2}{\eta \Delta}\left(\log\left(\gamma^{-1} n \epsilon^{-1}(1 - \tau(x_t))\right) - \eta \Delta \tau(x_t)\right) \\
&\quad + 2B(1 - \tau(x_t)) + t + 1 = W_t(x_t, \ldots, x_0) - 2\tau(x_t) + 1.
\end{aligned}
$$

Since $x_t \notin Z$, it follows that $2\tau(x_t) \geq 1$. Therefore,

$$
\mathbf{E}\left[W_{t+1}(x_t + \eta \tilde{A}_t x_t, x_t, \ldots, x_0\right] \leq W_t(x_t, \ldots, x_0).
$$

And so (3) holds in all cases.

The second rate supermartingale property (4) holds because, if success hasn't occurred by time $T < B$, then there are two possibilities: either the process hasn't stopped yet, or it stopped at a timestep where $x_t \in Z$. In the former case, by the non-negativity of the log function,

$$W_T(x_T, \ldots, x_0) = \frac{2}{\eta \Delta} \log \left( \gamma^{-1} n \epsilon^{-1} (1 - \tau(x_T)) \right) + 2B(1 - \tau(x_T)) + T \geq T.$$

In the latter case,

$$W_T(x_T, \ldots, x_0) = \frac{2}{\eta \Delta} \log \left( \gamma^{-1} n \epsilon^{-1} (1 - \tau(x_T)) \right) + 2B(1 - \tau(x_T)) + T$$

$$\geq B.$$

Therefore (4) holds.

We have now shown that $W_t$ is a rate supermartingale for Alecton. Next, we show that our bound on the initial value of the supermartingale holds. At time 0,

$$W_0(x_0) = \frac{2}{\eta \Delta} \log \left( \gamma^{-1} n \epsilon^{-1} (1 - \tau(x_0)) \right) + 2B(1 - \tau(x_0))$$

$$\leq \frac{2}{\eta \Delta} \log \left( \gamma^{-1} n \epsilon^{-1} \right) + 2B(1 - \tau(x_0))$$

$$= \frac{2}{\eta \Delta} \log \left( \frac{en}{\gamma \epsilon} \right) + 2B(1 - \tau(x_0)).$$

Therefore, applying Lemma 6,

$$\mathbf{E}\left[ W_0(x_0) \right] \leq \frac{2}{\eta \Delta} \log \left( \frac{en}{\gamma \epsilon} \right) + 2B \mathbf{E}\left[ 1 - \tau(x_0) \right]$$

$$\leq \frac{2}{\eta \Delta} \log \left( \frac{en}{\gamma \epsilon} \right) + B\sqrt{2\pi\gamma}.$$

This is the value given in the lemma.

Now, we show that $W_t$ is $(H, R, \xi)$-bounded. First, we give the $H$ bound. To do so, we first differentiate $\tau(x)$.

$$\nabla \tau(x) = \frac{2u_1 u_1^T x \left( (1 - \gamma n^{-1})(u_1^T x)^2 + \gamma n^{-1} \|x\|^2 \right) - 2(u_1^T x)^2 \left( (1 - \gamma n^{-1})u_1 u_1^T x + \gamma n^{-1} x \right)}{\left( (1 - \gamma n^{-1})(u_1^T x)^2 + \gamma n^{-1} \|x\|^2 \right)^2}$$

$$= \frac{2u_1 u_1^T x \gamma n^{-1} \|x\|^2 - 2(u_1^T x)^2 \gamma n^{-1} x}{\left( (1 - \gamma n^{-1})(u_1^T x)^2 + \gamma n^{-1} \|x\|^2 \right)^2}$$

$$= 2\gamma n^{-1} \frac{u_1 u_1^T x \|x\|^2 - x(u_1^T x)^2}{\left( (1 - \gamma n^{-1})(u_1^T x)^2 + \gamma n^{-1} \|x\|^2 \right)^2}.$$

Therefore,

$$\|\nabla \tau(x)\|^2 = 4\gamma^2 n^{-2} \frac{(u_1^T x)^2 \|x\|^4 - (u_1^T x)^4 \|x\|^2}{\left((1 - \gamma n^{-1})(u_1^T x)^2 + \gamma n^{-1} \|x\|^2\right)^4}$$

$$\leq 4\gamma^2 n^{-2} \frac{\|x\|^4 - (u_1^T x)^2 \|x\|^2}{\left((1 - \gamma n^{-1})(u_1^T x)^2 + \gamma n^{-1} \|x\|^2\right)^3}$$

$$\leq 4\gamma n^{-1} \frac{\|x\|^2 (1 - \tau(x))}{\left((1 - \gamma n^{-1})(u_1^T x)^2 + \gamma n^{-1} \|x\|^2\right)^2}$$

$$\leq \frac{4(1 - \tau(x))}{\left((1 - \gamma n^{-1})(u_1^T x)^2 + \gamma n^{-1} \|x\|^2\right)}$$

$$\leq \frac{4n(1 - \tau(x))}{\gamma \|x\|^2}.$$

Applying the assumption that $\|x\|^2 \geq 1$,

$$\|\nabla \tau(x)\| \leq \sqrt{\frac{4n(1 - \tau(x))}{\gamma}}.$$

Now, differentiating $W_t$ with respect to $\tau$ produces

$$\frac{dW}{d\tau} = -\frac{2n}{\eta \gamma \epsilon \Delta} \overset{'}{\log} \left(\gamma^{-1} n \epsilon^{-1}(1 - \tau)\right) - 2B.$$

So, it follows that

$$\|\nabla_x W_t(x, x_{t-1}, \ldots, x_0)\| \leq \left|\frac{dW}{d\tau}\right| \|\nabla \tau(x)\|$$

$$\leq \left(\frac{2n}{\eta \gamma \epsilon \Delta} \overset{'}{\log} \left(\gamma^{-1} n \epsilon^{-1}(1 - \tau)\right) + 2B\right) \sqrt{\frac{4n(1 - \tau(x))}{\gamma}}.$$

Applying our assumption that $\eta \gamma \epsilon \Delta B \leq 1$, it is clear that this function will be maximized when $\gamma^{-1} n \epsilon^{-1}(1 - \tau) = 1$. Therefore,

$$\|\nabla_x W_t(x, x_{t-1}, \ldots, x_0)\| \leq \left(\frac{2n}{\eta \gamma \epsilon \Delta} + 2B\right) 2\sqrt{\epsilon}$$

$$= \frac{8n}{\eta \gamma \Delta \sqrt{\epsilon}},$$

which is our given value for $H$.

Next, we give the $R$ bound. For Alecton, we have

$$\tilde{G}(x) = \eta \tilde{A} x = \eta n^2 e_i e_i^T A e_j e_j^T x.$$

Therefore,

$$
\begin{aligned}
\mathbf{E}\left[\left\|\tilde{G}(x) - \tilde{G}(y)\right\|\right] &= \eta n^2 \mathbf{E}\left[\left\|e_i e_i^T A e_j e_j^T (x-y)\right\|\right] \\
&= \eta n^2 \mathbf{E}\left[\left|e_i^T A e_j e_j^T (x-y)\right|\right] \\
&= \eta \sum_{i=1}^{n} \sum_{j=1}^{n} \left|e_i^T A e_j\right| \left|e_j^T (x-y)\right| \\
&= \eta \sum_{j=1}^{n} \left|e_j^T (x-y)\right| \left(\sum_{i=1}^{n} \left|e_i^T A e_j\right|\right) \\
&\leq \eta \sum_{j=1}^{n} \left|e_j^T (x-y)\right| \sqrt{n} \left(\sum_{i=1}^{n} (e_i^T A e_j)^2\right)^{\frac{1}{2}} \\
&= \eta \sum_{j=1}^{n} \left|e_j^T (x-y)\right| \sqrt{n} \left(e_j^T A^2 e_j\right)^{\frac{1}{2}} \\
&= \eta \sum_{j=1}^{n} \left|e_j^T (x-y)\right| \sqrt{n} \left(\sum_{k=1}^{\infty} \lambda_j^2 (u_k^T e_j)^2\right)^{\frac{1}{2}}.
\end{aligned}
$$

Applying the incoherence bound,

$$
\begin{aligned}
\mathbf{E}\left[\left\|\tilde{G}(x) - \tilde{G}(y)\right\|\right] &\leq \eta \sum_{j=1}^{n} \left|e_j^T (x-y)\right| \sqrt{n} \left(\sum_{k=1}^{\infty} \lambda_j^2 \mu^2 n^{-1}\right)^{\frac{1}{2}} \\
&= \eta \sum_{j=1}^{n} \left|e_j^T (x-y)\right| \sqrt{n} \left(\mu^2 n^{-1} \|A\|_F^2\right)^{\frac{1}{2}} \\
&= \eta \sum_{j=1}^{n} \left|e_j^T (x-y)\right| \mu \|A\|_F \\
&= \eta \mu \|A\|_F \|x - y\|_1.
\end{aligned}
$$

This agrees with our assignment of $R = \eta \mu \|A\|_F$.

Finally, we give our $\xi$ bound on the magnitude of the updates. By the same argument as above, we will have

$$
\begin{aligned}
\mathbf{E}\left[\left\|\tilde{G}(x)\right\|\right] &= \eta n^2 \mathbf{E}\left[\left\|e_i e_i^T A e_j e_j^T x\right\|\right] \\
&= \eta \mu \|A\|_F \|x\|_1.
\end{aligned}
$$

Applying the assumption that $\|x\|_1^2 \leq C$, produces the bound given in the lemma, $\xi = \eta \mu \|A\|_F C$. This completes the proof of the lemma. $\qquad \square$

Next, we prove the corollary that gives a bound on the failure probability of asynchronous Alecton.

*Proof of Corollary 3.* By Theorem 1, we know that for the constants defined in Lemma 2,

$$
P(F_T) \leq \frac{\mathbf{E}[W(0, x_0)]}{(1 - HR\xi\tau)T}.
$$

If we choose $B = T$ for the horizon in Lemma 2, and substitute in the given constants,

$$
\begin{aligned}
P(F_T) &\leq \left(\frac{2}{\eta\Delta} \log\left(\frac{en}{\gamma\epsilon}\right) + T\sqrt{2\pi\gamma}\right) \left(1 - \left(\frac{8n}{\eta\gamma\Delta\sqrt{\epsilon}}\right)(\eta\mu \|A\|_F)(\eta\mu \|A\|_F C)\tau\right)^{-1} T^{-1} \\
&= \left(\frac{2}{\eta\Delta T} \log\left(\frac{en}{\gamma\epsilon}\right) + \sqrt{2\pi\gamma}\right) \left(1 - \frac{8\eta n\mu^2 \|A\|_F^2 C\tau}{\gamma\Delta\sqrt{\epsilon}}\right)^{-1}.
\end{aligned}
$$

Now, for the given value of $\eta$, we will have

$$\frac{8\eta n \mu^2 \left\|A\right\|_F^2 C\tau}{\gamma \Delta \sqrt{\epsilon}} = \frac{\Delta \epsilon \gamma \vartheta}{2n\mu^4 \left\|A\right\|_F^2} \frac{8n\mu^2 \left\|A\right\|_F^2 C\tau}{\gamma \Delta \sqrt{\epsilon}}$$

$$= \frac{4C\vartheta\tau\sqrt{\epsilon}}{\mu^2}.$$

Also, for the given values of $\eta$ and $T$, we will have

$$\frac{2}{\eta \Delta T} \log\left(\frac{en}{\gamma\epsilon}\right) = \frac{2n\mu^4 \left\|A\right\|_F^2}{\Delta \epsilon \gamma \vartheta} \frac{\Delta^2 \epsilon \gamma \vartheta \sqrt{2\pi\gamma}}{4n\mu^4 \left\|A\right\|_F^2} \frac{2}{\Delta}$$

$$= \sqrt{2\pi\gamma}.$$

Substituting these results in produces

$$P(F_T) \le \sqrt{8\pi\gamma} \left(1 - \frac{4C\vartheta\tau\sqrt{\epsilon}}{\mu^2}\right)^{-1}$$

$$= \frac{\sqrt{8\pi\gamma}\mu^2}{\mu^2 - 4C\vartheta\tau\sqrt{\epsilon}},$$

which is the desired result. $\qquad\square$

# D    Simplified Convex Result

In this section, we provide a simplified proof for a result similar to our main result that only works in the convex case. This proof does not use any martingale results, and can therefore be considered more elementary than the proofs given above; however, it does not generalize to the non-convex case.

**Theorem 2.** *Under the conditions given in Section 3.1, for any $\epsilon > 0$, if for some $\vartheta \in (0,1)$ we choose constant step size*

$$\alpha = \frac{c\vartheta\epsilon}{2LM\tau\sqrt{\epsilon} + M^2},$$

*then there exists a timestep*

$$T \le \frac{2LM\tau\sqrt{\epsilon} + M^2}{c^2\vartheta\epsilon} \log\left(\frac{\left\|x_0 - x^*\right\|^2}{\epsilon}\right)$$

*such that*

$$\mathbf{E}\left[\left\|x_T - x^*\right\|^2\right] \le \epsilon.$$

*Proof.* Our goal is to bound the square-distance to the optimum by showing that it generally decreases at each timestep. We can show algebraically that

$$\left\|x_{t+1} - x^*\right\|^2 = \left\|x_t - x^*\right\|^2 - 2\alpha(x_t - x^*)^T \nabla \tilde{f}_t(x_t)$$

$$+ 2\alpha(x_t - x^*)^T \left(\nabla \tilde{f}_t(x_t) - \nabla \tilde{f}_t(\tilde{v}_t)\right) + \alpha^2 \left\|\nabla \tilde{f}_t(\tilde{v}_t)\right\|^2.$$

We can think of these terms as representing respectively: the current square-distance, the first-order change, the noise due to delayed updates, and the noise due to random sampling. Taking the expected

value given $\tilde{v}_t$ and applying Cauchy-Schwarz, (12), (13), and (14) produces

$$\mathbf{E}\left[\|x_{t+1} - x^*\|^2 \Big| \mathcal{F}_t, \tilde{v}_t\right] \leq \|x_t - x^*\|^2 - 2\alpha c \|x_t - x^*\|^2 + 2\alpha L \|x_t - x^*\| \|x_t - \tilde{v}_t\|_1 + \alpha^2 M^2$$

$$= (1 - 2\alpha c) \|x_t - x^*\|^2 + \alpha^2 M^2 + 2\alpha L \|x_t - x^*\| \sum_{i=1}^{n} \left| e_i^T x_t - e_i^T x_{t-\tilde{\tau}_{i,t}} \right|$$

$$\leq (1 - 2\alpha c) \|x_t - x^*\|^2 + \alpha^2 M^2$$

$$+ 2\alpha L \|x_t - x^*\| \sum_{i=1}^{n} \sum_{k=1}^{\tilde{\tau}_{i,t}} \left| e_i^T x_{t-k+1} - e_i^T x_{t-k} \right|.$$

We can now take the full expected value given the filtration, which produces

$$\mathbf{E}\left[\|x_{t+1} - x^*\|^2 \Big| \mathcal{F}_t\right] \leq (1 - 2\alpha c) \|x_t - x^*\|^2 + \alpha^2 M^2$$

$$+ 2\alpha L \|x_t - x^*\| \sum_{i=1}^{n} \sum_{k=1}^{\infty} P\left(\tilde{\tau}_{i,k} \geq k\right) \left| e_i^T x_{t-k+1} - e_i^T x_{t-k} \right|.$$

Applying (7) results in

$$\mathbf{E}\left[\|x_{t+1} - x^*\|^2 \Big| \mathcal{F}_t\right] \leq (1 - 2\alpha c) \|x_t - x^*\|^2 + \alpha^2 M^2$$

$$+ 2\alpha L \|x_t - x^*\| \sum_{i=1}^{n} \sum_{k=1}^{\infty} P\left(\tilde{\tau} \geq k\right) \left| e_i^T x_{t-k+1} - e_i^T x_{t-k} \right|$$

$$= (1 - 2\alpha c) \|x_t - x^*\|^2 + \alpha^2 M^2$$

$$+ 2\alpha L \|x_t - x^*\| \sum_{k=1}^{\infty} P\left(\tilde{\tau} \geq k\right) \|x_{t-k+1} - x_{t-k}\|_1,$$

and since only at most one entry of $x$ changes at each iteration,

$$\mathbf{E}\left[\|x_{t+1} - x^*\|^2 \Big| \mathcal{F}_t\right] \leq (1 - 2\alpha c) \|x_t - x^*\|^2 + \alpha^2 M^2$$

$$+ 2\alpha L \sum_{k=1}^{\infty} P\left(\tilde{\tau} \geq k\right) \|x_t - x^*\| \|x_{t-k+1} - x_{t-k}\|.$$

Finally, taking the full expected value, and applying Cauchy-Schwarz again,

$$\mathbf{E}\left[\|x_{t+1} - x^*\|^2\right] \leq (1 - 2\alpha c)\mathbf{E}\left[\|x_t - x^*\|^2\right] + \alpha^2 M^2$$

$$+ 2\alpha L \sum_{k=1}^{\infty} P\left(\tilde{\tau} \geq k\right) \sqrt{\mathbf{E}\left[\|x_t - x^*\|^2\right] \mathbf{E}\left[\|x_{t-k+1} - x_{t-k}\|^2\right]}.$$

Noticing that, from (14),

$$\mathbf{E}\left[\|x_{t-k+1} - x_{t-k}\|^2\right] = \mathbf{E}\left[\left\|\alpha \tilde{G}(\tilde{v}_{t-k})\right\|^2\right] \leq \alpha^2 M,$$

if we let $J_t = \mathbf{E}\left[\|x_t - x^*\|^2\right]$, we get

$$J_{t+1} \leq (1 - 2\alpha c)J_t + \alpha^2 M^2 + 2\alpha^2 LM \sum_{k=1}^{\infty} P\left(\tilde{\tau} \geq k\right) \sqrt{J_t}$$

$$= (1 - 2\alpha c)J_t + \alpha^2 M^2 + 2\alpha^2 LM\tau\sqrt{J_t}.$$

For any $\epsilon > 0$, as long as $J_t \geq \epsilon$,

$$\log J_{t+1} \leq \log J_t + \log\left(1 - 2\alpha c + \alpha^2 M^2 \epsilon^{-1} + 2\alpha^2 LM\tau\epsilon^{-\frac{1}{2}}\right)$$
$$< \log J_t - 2\alpha c + \alpha^2 M^2 \epsilon^{-1} + 2\alpha^2 LM\tau\epsilon^{-\frac{1}{2}}.$$

If we substitute the value of $\alpha$ chosen in the theorem statement, then

$$\log J_{t+1} < \log J_t - \frac{c^2 \vartheta \epsilon}{2LM\tau\sqrt{\epsilon} + M^2}.$$

Therefore, for any $T$, if $J_T \geq \epsilon$ for all $t < T$,

$$T < \frac{2LM\tau\sqrt{\epsilon} + M^2}{c^2 \vartheta \epsilon} \log\left(\frac{J_0}{J_T}\right),$$

which proves the theorem. $\square$