[Reviews · NeurIPS 2015]

Submitted by Assigned_Reviewer_1

This paper presented a unified theoretical framework for analyzing the convergence rates of asynchronous and low-precision random algorithms like stochastic gradient descent (SGD). The basic idea is to use the martingale-based analysis which can be used to capture the rich noise models arising from random algorithms. Moreover, a new SGD method called BUCKWILD! is proposed to use lower-precision arithmetic. Experiments on real-world datasets are used to evaluate the performance of the proposed method.

The martingale-based unified analysis seems to be interesting. It provides another tool to analyze some existing methods, such as HOGWILD!.

However, the analysis of this paper provides no hint to improve the algorithm of HOGWILD!, and the convergent rate in this paper is only the same as that in HOGWILD!. This makes this paper not very promising. Furthermore, the speedup of BUCKWILD! is not satisfactory. For example, the speedup of 8-bit int with 24 threads is only 6, which is far less than the ideal speedup which is 24.

Summary: The martingale-based unified analysis seems to be interesting. However, the analysis of this paper provides no hint to improve the algorithm and convergence rate of the existing algorithm HOGWILD!.

Submitted by Assigned_Reviewer_2

This paper is an essentially theoretical contribution regarding convergence rates for the so-called "Hogwild"-style algorithms for stochastic gradient descent.

In these algorithms, the gradient step is produces asynchronously over different chunks of the dataset in parallel, with results updating current weights as they are completed, independent of other parallel updates.

Previously, demonstrating theoretical convergence has been difficult and somewhat brittle.

In this paper, the authors present an analysis framework that is able to extend convergence results to a variety of specializations/generalizations of the algorithm.

They show that one of their proven variants, "Buckwild" provides significant real-world speedups by using lower precision arithmetic to compute the gradient steps.

As far as the paper goes, it is generally good.

I had little trouble reading and understanding the paper (I think), and they make a point to explain the maths in an intuitive fashion, insofar as it is possible.

The authors do a good job placing their work in the context of other research.

I admit that I'm not an expert in convergence proofs of this type, so I'll have to leave it to others to do a robust check of the correctness thereof.

If the proof is correct, it seems to satisfy the usual criteria for a reasonable proof of this type (part of which is that it is by itself not sufficient to show the practical usefulness of the underlying algorithm).

One concern I have is that it's not obvious where the work of others ends and the work done by the authors begins.

All of the citations are there, I'm just not sure how closely related previous results are to what's being show here.

For example, on p1, 51 the authors mention at least one other version of SGD that uses reduced precision arithmetic, but no details from the paper are referenced in section 3.2.

Is Buckwild closely related?

Are the results similar?

Moreover, I could swear I remember a paper in which supermartingale methods are use to prove convergence results in some Hogwild-related paper, but neither Google nor memory is serving me well, so I'll take the authors word that they're first to this idea as they seem to claim (in any case, they should please give the literature one last look to make sure that this is so).

Another concern is the broadness of the result:

I'm not sure how popular fixed point arithmetic is in this area, especially considering the dominance of GPU calculations in this area.

The result for the non-convex optimizations is nice, but we decide the fixed point result isn't that important, it starts to seem a bit thin to me.

Of course, you often have to do limited precision calculations to use the GPU (e.g., 64 -> 32 bit floats).

Do these results apply in that case?

Might be a nice addition.

The empirical results are somewhat lightweight, but it's to be expected in a mostly theoretical paper.

I'd like to see confidence bars on the speed comparison, and I'd like to see every thread size from 1 to 24 if they can manage it.

Why does the 32-bit version slow down at high numbers of threads?

Figure 1b. seems unnecessary to me:

I'm willing to take the authors word for it that the two algorithms converge to roughly the same answers in roughly the same fashion.

Table 1 is kind of a fun result by itself which I'd love to see explored more, maybe in a different paper.

It seems crazy to me that we can throw out a whole bunch of information and have virtually no impact on performance.

Honestly, let's just use a single bit and see what happens.

Could you reference other papers that show something similar here?

Incidentally, Googling related terms brings up this paper both at arvix and at the author's homepage.

Is this what we're doing now?

At a minimum, it makes blind review kind of meaningless.
Summary: An theoretical analysis of Hogwild that accounts for common approximations used by current literature.

A decent paper, with some reasonable experimental results.

Submitted by Assigned_Reviewer_3

Positives: * unified analysis of asynchronous update algorithms. Asynchronous algorithms are fast and empirically relevant and analyzing their convergence rates/properties is definitely useful to the community * presents convergence analysis of Hogwild style algorithms without requirement of sparsity assumptions * their analysis for asynchronous SGD style algorithms also extends to (a specific) non-convex case, which is a big plus * a low-precision algorithm, which I believe, are getting popular in machine learning community and a theoretical framework to analyze them would indeed be interesting

Concerns: * the limited precision is only useful when features space is not large (I am assuming that since the input is fixed precision, the feature weights that are updated based on input data is also represented via low precision). But in many industry-scale real world problems, the limited precision might lead to feature collisions (due to hashing into a limited feature space). It would be interesting to observe the robustness of low-precision arithmetic for datasets with millions of features * I would temper down the claims made in end of section 4 about the low-precision asynchronous updates leading to 2.3x speedup without any significant error degradation. Surely this applies to small subset of datasets that were empirically analyzed in the paper.
Summary: Overall, I think is an interesting paper that proposed an unified analysis of asynchronous SGD type algorithms. What more useful is that their analysis extends to specific non-convex use cases as well as fixed-point algorithms and probably a combination of both. The empirical results for the low-precision case and the claims made based on those are a bit preliminary and merits more exhaustive treatment. It would be really interesting to see how the accuracy varies and what other tradeoffs come into play when using low-precision arithmetic for very high dimensional datasets (features ~ a few million, eg. Criteo click-prediction dataset).

Submitted by Assigned_Reviewer_4

Minor comments -------------------

line 320: "we focus on the rank one version of the problem in (16)": actually, (16) is already the rank-one problem. maybe the authors meant to put the full rank in (16)?
Summary: The paper is clearly written, considers an interesting topic and unifies the analysis of asynchronous updates with quantization.

Author Feedback
Author rebuttal: We thank all the reviewers for their reviews and feedback, which have helped us improve our paper. Two important points mentioned by the reviewers deserve specific comment:

1. In the introduction, R2 mentions that it is "not obvious where the work of others ends and the work done by the authors begins." We agree that this could be made more clear, and we will add prose in the introduction that delineates cited work from ours. For the example mentioned by R2, their work on low-precision algorithms focuses specifically on an empirical analysis of neural network training and provides no theoretical guarantees, which differs from our work which gives theoretical guarantees for a general class of problems.

2. Both R1 and R2 had questions about the experimental section, particularly the results given in Figure 1(a). R1 says "the speedup of BUCKWILD! is not satisfactory. For example, the speedup of 8-bit int...is far less than the ideal speedup." R2 has a similar question: "Why does the 32-bit version slow down at high numbers of threads?"

We don't get linear speedup because we are bound by the memory bandwidth available on the core. The 32-bit version slows down at 24 threads because there are only 12 physical cores and the whole workload is already memory-bandwidth bound at 12 cores. Adding 2x logical cores with hyperthreading provides no benefits, while trashing the L1 & L2 caches and increasing conflicts from updating the model concurrently. However, for the same number of threads, limited-precision achieves higher performance than full-precision, both because it decreases the memory load and because we are able to compute with high-throughput SIMD instructions. If an algorithmic change makes HOGWILD! run faster, we expect BUCKWILD! to also run faster. We will amend the paper to make clear how our experiments are affected by the combination of SIMD, memory bandwidth, and caching.

Reviewer 1

R1 notes that "convergent rate in this paper is only the same as that in HOGWILD!" While this is true when measured in terms of number of updates, our low-precision BUCKWILD! updates each take less time than HOGWILD! steps, so BUCKWILD! executes in less wall clock time. We will modify Section 3.2 to include this point. Also note that, compared to previous results, our rate applies under more general conditions.

R1 states that we give no "hint to improve the algorithm of HOGWILD!" We will amend the paper to make it more clear that low-precision BUCKWILD! updates are intended to be used as such an improvement in many cases.

Reviewer 2

R2 mentions remembering a HOGWILD!-related paper "in which supermartingale methods are use to prove convergence." We are unaware of any such paper, but if we can identify such a paper, we will cite it properly.

R2 also mentions that they are "not sure how popular fixed point arithmetic is in this area," and is concerned about broadness and running on GPUs. While we agree that the performance depends on the instruction set of a device, GPUs do typically have 16-bit float, and 8- through 64-bit integer types (see "Accelerating GPU computation through mixed-precision methods"). On any device, we can always save on memory bandwidth by using low-precision arithmetic, and our results do still apply for even 32-bit floating point operations on GPUs. Additionally, people often run SGD on the CPU in practice, so it is important to understand how we can speed it up.

We will modify Figure 1a to include more thread counts.

R2 also is interested in algorithms that use just a single bit. There are reasons to expect that this will work well, including the ease of doing 1-bit multiplies and other bitwise operations on the CPU. Unfortunately, our theoretical bounds get really poor as the number of bits goes to 1, possibly because we end up discarding too much information. We ran 1-bit and 4-bit experiments, and we will amend the paper to briefly include our results. For the 4-bit case, because Intel CPU does not have hardware-level support for SIMD with half-byte data, we observe it runs slower than 8-bit due to extra computation cost.

Reviewer 3

We thank R3 for their review, and will correct the mentioned issue.

Reviewer 4

R4 notes that "the limited precision is only useful when features space is not large." It is true that the algorithm will run slower with a larger feature space. However, compared to a full-precision version, each low-precision update will still run faster even for a large feature space. As future work, we will test how the error due to low-precision updates interacts with feature collisions, but such an analysis is beyond the scope of this paper.

We will be more specific with our claims in Section 4 about the conditions under which we achieve the stated speedup without significant errors.

Reviewer 5

We thank R5 for their review.

Reviewer 6

We thank R6 for their review.